# DRL-based low Latency control for Endoscopic operations

Abdelrahman Soliman*, Hoda Helmy*, Elias Yaacoub*,
Mohamed Mabrok**, Nikhil V. Navkar‡, and Amr Mohamed*
*Department of Computer Science and Engineering, Qatar University, Qatar
**Department of Mathematics, Statistics, and Physics, Qatar University, Qatar
‡Department of Surgery, Hamad Medical Corporation, Qatar
Email: {as1701600,hoda.helmy, m.a.mabrok}@qu.edu.qa,
{eliasy, amrm}@ieee.org, nnavkar@hamad.qa

*Abstract*—Minimally invasive surgery has seen significant advancements with the introduction of robotic systems, which are highly desirable due to their ability to enhance treatment scalability and precision. This study aims to develop an effective and intelligent system for controlling and streaming the endoscope camera during endoscopic operations. The proposed system leverages head motion data from the HoloLens inertial measurement unit (IMU) to control the endoscope camera's robotic arm. Additionally, a Deep Reinforcement Learning (DRL) technique is employed to manage the region of interest adaptively (ROI), thereby mitigating wireless channel impairments in the operating room and improving the surgeon's interaction and quality of experience (QoE). We developed a proof-of-concept by interfacing the HoloLens with the Gazebo simulation environment for robotic arm control. The DRL model demonstrated its efficacy by intelligently reducing the communication delay and enhancing image quality by incorporating machine learning techniques. Specifically, the DRL model reduced delay by 12.56% and increased image quality by 26.6%. Furthermore, a fixed-frame technique resulted in an additional delay reduction of 18%. The successful establishment of the proof-of-concept and the comprehensive analysis of the findings underscore the potential impact of our contribution to advancing intelligent, efficient, and easily controllable endoscopic surgical procedures.

*Index Terms*—Deep reinforcement learning, Assistive Control, Augmented reality, minimally invasive surgery, endoscopy, Simulation.

## I. INTRODUCTION

Endoscopy is a medical imaging technique that allows doctors to examine internal organs and cavities using an endoscope. Initially, it was primarily utilized to diagnose gastrointestinal issues, but subsequently, it has been adopted for performing minimally invasive surgeries (MIS). A flexible endoscope is a long, thin, flexible tube with a light and a camera attached to its tip that can be inserted inside a patient's body to access the internal organs [1]. The surgeon can visually examine a concerned area by obtaining images on a video monitor.

During an endoscopy procedure, it is essential to have an assistant who can effectively manage and position the endoscope as per the surgeon's directions. The assistant's involvement plays a vital role in the procedure's success. Nonetheless, miscommunication between the assistant and the surgeon can cause challenges and potentially influence the procedure's results [2]. Integrating robot arms into endoscope operations enhances precision and control, allowing surgeons to manipulate the endoscope camera directly. This transfer of control can be further streamlined through augmented reality (AR) technology, which can provide real-time, intuitive guidance, thus simplifying the camera manipulation process and reducing the dependency on assistants.

The medical sector has recently widely adopted virtual and augmented reality devices for surgical training. However, the benefits of augmented reality devices go beyond training and can extend to other areas, such as video endoscopy monitoring. The Inertial Measurement Unit (IMU) of a head-mounted device, for example, can be leveraged to provide better control over the position of the endoscope [3] [4].

To enhance the surgeon's experience, we directed our efforts toward improving the quality of the video feed to provide a more satisfactory real-time experience. The motivation for seeking a balance between video quality and delay stems from challenges inherent in wireless communication channels, especially in operating rooms where numerous electronic devices can cause interference. Despite dedicated connections, wireless channels are susceptible to fluctuating conditions

like multipath effects and electromagnetic interference, potentially leading to increased delays and packet loss. Thus, optimizing the video frame quality while managing delay becomes crucial to ensuring real-time transmission and maintaining high-quality imaging during endoscopic procedures.

Our proposal involves using an adaptive Region of Interest (ROI) detection system and an intelligent Deep Reinforcement Learning (DRL) model to balance the overall quality of the video frame with the amount of delay it incurs.

We employ a higher-quality ROI than the background (non-ROI) and dynamically adjust ROI size based on network circumstances. The surgeon can make educated selections with high-quality ROI displays. With the suggested approach, its size will vary dynamically based on network conditions. The non-ROI zone can be presented at reduced quality, requiring lower network data rates. Higher data rates for higher ROI quality cause longer transmission delays, especially on networks with low capacity. Thus, DRL chooses the smallest ROI size and lowest non-ROI quality when network circumstances are poor and the converse when they are excellent. This method reduces endoscopic delays and improves video transmission latency without affecting ROI.

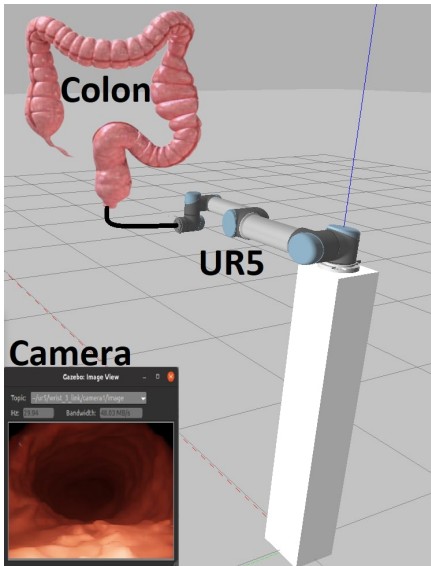

Fig. 1: Gazebo simulation and navigation through human body

The contributions of this paper can be summarized as follows:

- We provide an enhanced adaptive ROI DRL solution for more responsiveness and low latency in the wireless stream for the HoloLens.

- We develop a proof-of-concept for the whole control system, including the simulation environment for the endoscope robotic arm. And the interfacing for the HoloLens head motion control.
- Our proposal extends to an integrated approach, seamlessly combining the DRL solution with the HoloLens head motion control to achieve a cohesive and efficient system.

The structure of this paper is as follows: Section II provides an overview of related work; Section III outlines the system architecture, and Section IV describes how the system was constructed in the simulation environment, along with the DRL approach. Section V discusses the DRL results in this work and showcases the simulation results. Finally, Section VI concludes the paper and outlines future research directions.

*A. Related Work*

Reducing the latency or delay of video streams has been a critical area of research in recent years, and many solutions have been suggested. These techniques include adaptive bit rate streaming, edge computing, and network optimization. Adaptive bit rate streaming has been a popular approach for reducing the delay of video streams. Work in [5] proposed an adaptive transmission technique for a delay-constrained wireless video that dynamically adjusts the transmission rate based on the buffer occupancy and the delay requirements of the video. For video encoding, they also use adaptive slice partitioning and intra-refreshment. Simulations and experiments are used to test the proposed method, which shows that it can get better video quality with less delay than existing methods. Similarly, [6] introduced an adaptive transmission rate control algorithm that models the problem of video rate control over a time-varying channel as a Markov decision process (MDP). They generated two algorithms, one based on a greedy approach and the other on a stochastic gradient descent (SGD) method. These publications are similar to ours because our DRL model seeks to adaptively choose ROI size and non-ROI quality based on network conditions. However, these methods only take action depending on the current conditions, while our intelligent solution takes action depending on the expected throughput after analyzing the history of network traces.

Edge computing has also been explored as a solution to lower the latency of video streams. Work in [7], [8] proposed an edge computing-based framework that offloads the processing tasks to edge servers, reducing the latency and improving the user experience. They

found that their proposed framework could significantly improve system performance compared to the standard Dynamic Adaptive Streaming over HTTP (DASH)-based system with fixed bitrates for transcoding. This is because their proposed framework's method for transcoding is more efficient, and the bandwidth adjustment can make better use of the bandwidth. Additionally, authors in [9] presented a model that utilizes edge computing to enhance video quality and reduce the delay in video transmission. They also concentrated on the possibilities and effects of cutting-edge technologies on multimedia. The authors concluded that cutting-edge technologies could help solve many problems with interactive media and video streaming.

Researchers have explored methods to minimize video stream delays, such as an intelligent algorithm for wireless networks. According to [10], this algorithm adjusts the quantization levels of video frames in accordance with packet loss rates and quality of experience (QoE) standards. An application model framework was also developed to automatically evaluate QoE, adjusting the quantization parameter (QP) based on detected network packet loss to optimize end-user QoE. Despite these advancements, a gap exists in applying machine learning combined with network condition monitoring to effectively reduce video streaming delays, particularly in the healthcare sector.

Surgeons have increasingly opted for AR and VR technologies in robotic surgeries. These technologies allow surgeons to view complex anatomical structures and surgical procedures in real-time, providing an immersive experience that can improve precision and accuracy. In [11], they utilized simulators based on VR technology to enable trainees to interact with virtual organs and tissues using the same surgical tool handles used in actual minimally invasive surgery (MIS). During the training, trainees could view images of tool-tissue interactions on a monitor, providing a realistic experience that closely imitated actual laparoscopic procedures. VR simulators allowed the trainees to have a tactile and interactive experience, which could enhance their skills and proficiency in a safe and controlled environment. Similarly, [12] combined an AI module with AR technology to create trajectories for surgical procedures. The AI module can use reinforcement learning to learn how to perform operations and produce trajectories according to the current task. To make the learning process more intuitive and immersive, the trajectory plan is projected onto a stereo video, which provides trainees with intelligent guidelines in 3D AR.

While previous methods significantly reduce human assistants' need for manual adjustments, they do not eliminate it. Our project builds on this work by using two-way wireless communication, leveraging the Hololens for IMU data to move the endoscope camera, and streaming the video directly to the head-mounted display (HMD). The key difference is our integration of intelligent machine learning algorithms and network status tracking to minimize video stream delay, a novel contribution not addressed in previous research.

## II. SYSTEM ARCHITECTURE

### A. System model

The endoscopy field presents surgeons with many challenges, including efficiently controlling the endoscope camera without needing an in-room assistant. To address this challenge, we have developed an innovative system utilizing augmented reality devices, empowering surgeons to manipulate the view of the endoscope camera according to their desired perspective. Another significant hurdle is the impact of wireless channel variability in the operating room, which can lead to data transmission delays and degrade the quality of the endoscope video stream. To tackle this issue, we introduce a deep reinforcement learning (DRL) based model that intelligently reduces delay. This model dynamically adjusts the size of the Region of Interest (ROI) in the video frame to be displayed at the highest quality while optimizing the quality of the non-ROI region based on the anticipated throughput. By tackling these obstacles, our research aims to enhance the precision and efficiency of endoscopy procedures, fostering improved surgical outcomes and patient care.

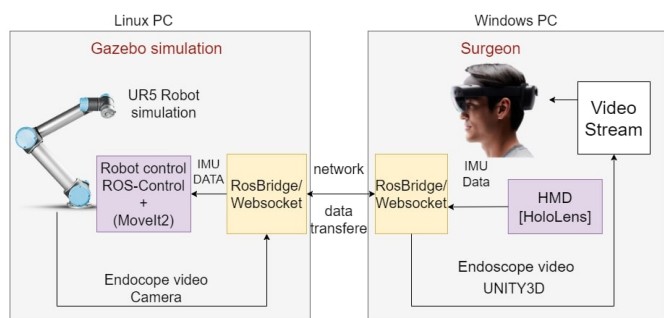

Fig. 2: System Model Overview.

Our system architecture comprises multiple integral components, including the HoloLens, a robotic holder that supports the endoscope camera in a simulated environment, a computer processing unit, and specialized software such as Unity3D, GAZEBO (running

on ROS2), RVIZ, and Moveit libraries. During the endoscopic procedure, the physician wears the head-mounted display (HMD) or HoloLens to visualize augmented images and provides input through the integrated inertial measurement unit (IMU) to navigate and manipulate the endoscope.

The computer processing unit assumes a critical role in the system, as it receives images obtained from the endoscope camera and facilitates their transmission to the HoloLens. Simultaneously, it tracks the head movements of the physician, ensuring synchronized visual feedback. Through intuitive head tilting, the physician can adjust the endoscope's orientation, enabling effortless navigation within the patient's body and augmenting maneuverability throughout the procedure.

In Figure 2, the system architecture illustrates integrating two distinct operating systems (OS) on separate devices, essential for establishing communication between remote systems via the ROS bridge. This model incorporates two simulation engines, GAZEBO and Unity3D, which are pivotal in the system's development and functionality. The decision to utilize separate machines rather than alternative approaches like Windows Subsystem for Linux (WSL) or a Linux virtual machine is justified by the need for dedicated hardware resources and real-time performance that these simulations demand. By distributing tasks across distinct OS environments, our approach optimizes system responsiveness and effectively mitigates latency in endoscopic video streaming.

### B. Virtual Simulation

*1) GAZEBO and the Simulated World:* Our project utilizes Gazebo11, a 3D open-source simulator, to create realistic scenarios for testing the Universal Robot 5 (UR5), which is equipped with a camera and six degrees of freedom. The UR5, modeled with a universally accessible URDF file, is controlled via the MoveIt2 framework configured in ROS2 Foxy on Ubuntu Linux - Focal Fossa (20.4). This setup enables precise motion planning and manipulation within the virtual environment, ensuring safe testing before real-world application. Gazebo11, integrated with ROS2, enhances the simulation with realistic physics and interactive plugins, while MoveIt2 provides comprehensive kinematics and trajectory management tools.

*2) Unity3D and ROSBridge Integration:* A surgeon uses a Unity3D-configured head-mounted display (HMD) to control a UR5 robotic arm as part of the surgical system. Unity3D is responsible for displaying

the output from the endoscopic camera in real-time within the Gazebo simulation environment. This setup uses the inertial measurement unit (IMU) data from the HMD to dynamically adjust the camera's perspective to align with the surgeon's head movements. ROS Bridge serves as the crucial communication link in this configuration, managing the bidirectional exchange of IMU data between the HMD and the UR5 robot and streaming camera data back to the surgeon's HMD. As shown in Figure 3, this integration facilitates seamless interaction between the Unity3D environment on Windows and the ROS environments running on Linux via TCP/IP, ensuring efficient data flow. The system precisely controls the robotic arm's six degrees of freedom (DOF) to move the end effector to the surgeon's view, eliminating the need for manual adjustments. This is made possible by the MoveIt2 library and ROS's inverse kinematics.

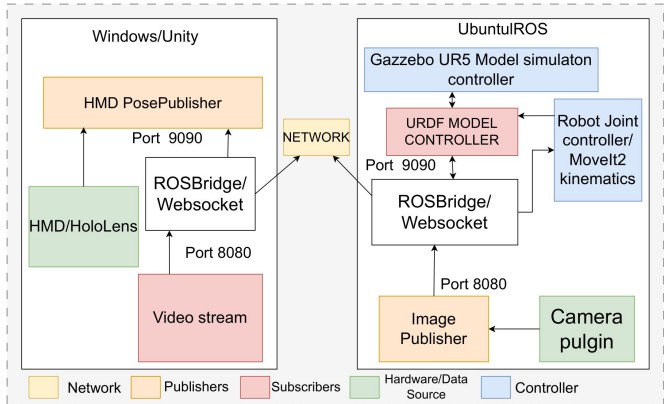

Fig. 3: Data flow in the system.

### C. Deep Reinforcement Learning

After defining the system model and presenting the simulation architecture and the data flow, we will discuss an intelligent approach to reduce the stream delay. Leveraging machine learning approaches to reduce delay is significant because it helps us better comprehend the complexities and hidden correlations between data and outcomes. Our preference for employing DRL stems from the requirement for a sophisticated decision-making model that considers the previous, current, and following possible states; one could be the throughput. DRL comprises several constituent elements that will be elaborated upon in the subsequent subsections.

*1) Environment Design:* The environment is where we carry out actions and obtain states or observations. In our case, the environment consists of the video

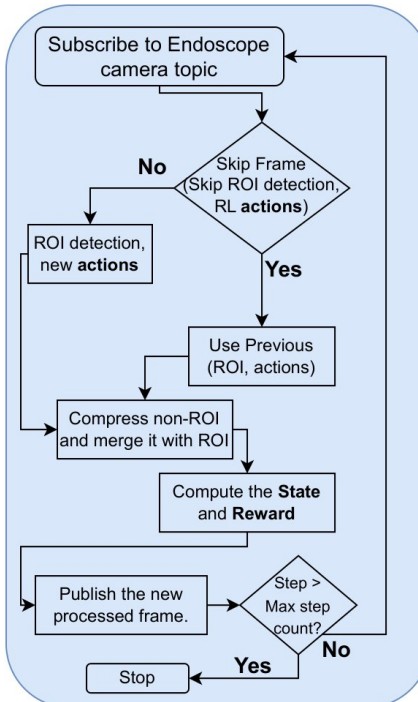

Fig. 4: RL flowchart.

frames captured by the camera on our simulated robot. As shown in Figure 4, the first step is to grab the frame by subscribing to the endoscope camera topic. Then, the agent performs actions depending on the value of the Boolean skip frame. If true, it will skip the original ROI detection and use the previous actions. When false, it will use the customized shallow convolutional neural network (S-CNN) to get the original ROI and then apply the new actions. After getting the ROI and the actions, we start compressing the non-ROI region and merging it with the ROI to get the final frame. To evaluate the actions taken, we compute the observations and reward. Finally, the final frame is published to a new topic that Unity will use for streaming to the HoloLens.

*2) Action Spaces and States:* Action space consists of three continuous parameters, each of which is normalized between zero and one to assist the model in enhancing its training while also avoiding divergence. The action space is as follows: A=$\{X, Y, QF\}$, where $QF$ is the quality factor, and $X$ and $Y$ are the measures of how much the original width and length of the ROI should increase.

Each state (s) in the state set S consists of delay, quality, and throughput. Hence, $s=\{\gamma, \lambda, T\}$, where ($\gamma$) is the total delay, which is the sum of the communication

delay and the processing delay. It is given by:

$$\gamma = (S(x,y,q)/throughput) + (ProcessingDelay) \tag{1}$$

($\lambda$) is the quality, and ($T$) is the throughput. Equation (1) is used to determine the communication delay. In the numerator, we calculate the total size of the frame in bytes using the regression model shown in Equation (2),

$$S(x,y,q) = p_{00} + p_{10} \cdot x + p_{01} \cdot y + p_{20} \cdot x^2 + p_{11} \cdot x \cdot y$$
$$+ p_{02} \cdot y^2 + p_{30} \cdot x^3 + p_{21} \cdot x^2 \cdot y + p_{12} \cdot x \cdot y^2 + p_{03} \cdot y^3 \tag{2}$$

In (2), the following are the values for the regression model's parameters that were obtained using the MATLAB regression tool with an R-Square of 0.822:
$p_{00} = 6.256 \times 10^4$, $p_{10} = -0.2356$, $p_{01} = 432.4$, $p_{20} = 1.412 \times 10^{-6}$, $p_{11} = 0.001398$, $p_{02} = -8.561$, $p_{30} = -2.637 \times 10^{-12}$, $p_{21} = -8.87 \times 10^{-9}$, $p_{12} = 6.147 \times 10^{-6}$, $p_{03} = 0.04034$.

The regression model's inputs are the original ROI width + action (X), original ROI height + action (Y), and quality factor action (QF). Then, to obtain the transmission delay, we divide the frame size by the throughput in the first part of Equation (1). The throughput is highly variable and dynamic, especially in surgery rooms, where reliable and efficient communication is critical to the success of procedures and where changes in throughput can significantly affect network performance. However, throughput can vary depending on network congestion, interference, distance, and network setup. Physical barriers such as walls, floors, and medical equipment can block or weaken the wireless signal, leading to interference and decreased network performance. Other wireless devices in the surgery room, such as monitoring equipment, can also interfere with the wireless signal and cause slow or dropped connections, further exacerbating the variability in throughput. Additionally, high volumes of network traffic, especially during critical surgical procedures, can cause congestion and slow down the network, leading to more variability in throughput. The quality and capabilities of the network equipment, including access points and antennas, must be of high quality to guarantee stable and efficient connections over the wireless network's frequency band and minimize variability in throughput. We measured the network traces between the client and server, which are the HoloLens and the stream computer, using the

Ookla speed test framework [13]. This test was run multiple times at various periods of the day to simulate different network congestion scenarios.

*3) Reward Function:* The reward function in equation (3) demonstrates how it directs and controls the model's actions to help it achieve specific objectives. Any deviation from the stated goals of our reward function, i.e., minimizing delay and maximizing quality, is considered a negative reward, i.e., a penalty. In our setup, we discovered that throughput values less than 103,076 MB/s result in unacceptable delays. So, we have used this value as a throughput reference point to inform the model that the throughput below this point is poor. Additionally, we added that a negative reward should be given if the sum of the actions is below or equal to 0.2 when throughput is low and greater than or equal to 0.9 when throughput is high.

$$\textbf{Reward}(\gamma, \lambda, T, A) := \begin{cases} (\gamma^{-1} + \lambda^{-1}) & \text{if } T < 103,076 \, \& \, \Sigma A \leq 0.2 \\ (\gamma + \lambda) & \text{if } T \geq 103,076 \, \& \, \Sigma A \geq 0.9 \\ -1 & \text{else} \end{cases}$$

$$(3)$$

*4) Deep Reinforcement Learning Algorithm:* The agent aims to find the policy that maximizes the overall discounted reward. Since basic RL algorithms cannot handle continuous actions, we can utilize a Deep Deterministic Policy Gradient (DDPG) or Soft Actor-Critic (SAC). The SAC algorithm was chosen because it was stable in our context and produced superior results. A further factor is that authors in [14] concluded that even with some continuous action spaces that are not particularly complicated, DDPG works well. Still, it does not have a rapid convergence rate. It is well known that SAC can train and test a stochastic policy using entropy regularization and in an off-policy manner. Moreover, it uses a separate network for policy and one for value functions, following an actor-critic design.

## III. RESULTS AND DISCUSSION

This section presents a detailed analysis of the training results of the Deep Reinforcement Learning (DRL) model. We also investigate the impact of model implementation on latency and quality, highlighting the new changes and improvements that have been achieved. Furthermore, we compare our suggested solution's performance with the baseline and benchmark solutions. Lastly, we propose a novel solution to reduce processing time.[1]

[1]A demo video can be accessed at https://youtu.be/E-_7eFx1ekA.

### A. S-CNN Results

The S-CNN model was trained using Python and the TensorFlow library, a widely adopted deep learning framework. The Adam optimizer, known for its robustness and fast convergence rate, was employed with default settings during the training phase. An endoscope frame dataset was utilized to facilitate the training process. This dataset consists of 1540 frames and an expert in the field of endoscopy manually labelled each frame's ROI. To augment the data, various image augmentation techniques, such as crop, rotation, and filtering, were utilized to enhance the variability of the data while preventing over-fitting. The final dataset comprised over 30,000 frames, significantly more than the original sample size. The evaluation of the model's performance indicates that the train and test accuracy scores are notably high. Consequently, the model exhibits a remarkable ability to identify the ROI accurately.

### B. DRL training Convergence

After 15,000 timesteps of training at a learning rate of 0.002, as shown in Figure 5, the average episode reward converged to around 56.

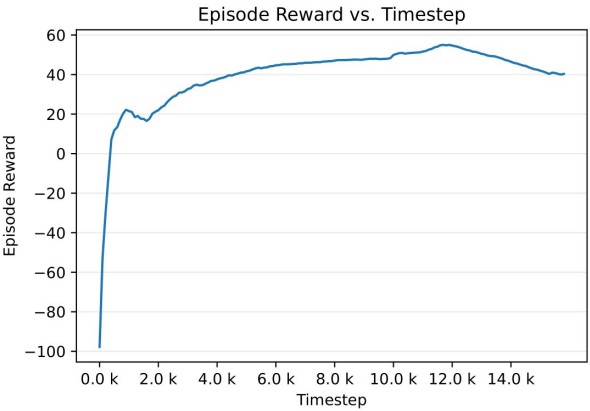

Fig. 5: Episode convergence for DRL training

### C. Comparative study with the Baseline and Benchmark

Comparing the DRL technique to a baseline and Benchmark is necessary to assess its performance. The Baseline scenario has the lowest ROI and non-ROI quality, while the Benchmark scenario has the best possible ROI size and non-ROI quality. The image quality is measured using the peak signal-to-noise ratio (PSNR) metric. As shown in Figure 6, experiment results show

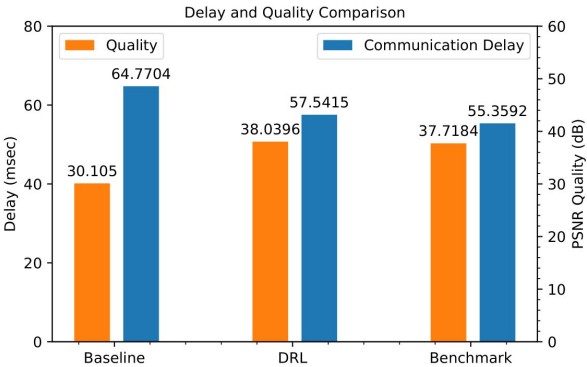

Fig. 6: Comparative study with the Baseline and Benchmark

that the DRL algorithm significantly improves PSNR quality, with a score of approximately 26.6% higher than the baseline (38.04 compared to 30.11). Also, the communication delay is close to the benchmark value.

Overall, the DRL algorithm offers promising results for both total delay and PSNR quality. It significantly improved PSNR quality by 26.6% over the baseline while decreasing the communication delay by 12.56%. These findings suggest that the DRL algorithm may be suitable for reducing delay while improving image quality.

### D. Optimizing processing time

In our study, despite access to high computational resources and a primary focus on minimizing communication delays, we introduced a technique to optimize processing time for low-cost, low-latency video streaming. We proposed skipping the region of interest (ROI) prediction in consecutive frames due to their significant similarity and slow characteristic changes. This approach was validated using the Structural Similarity Index (SSIM), where thirty consecutive frames yielded a high average SSIM score of 0.951, confirming frame correlation. As shown in Figure 7a, skipping frames significantly reduces total delay, and Figure 7b demonstrates that this does not adversely affect the quality. We determined the optimal skipping interval by comparing delays under fixed and dynamic throughput scenarios. We found that the total delay stabilizes after skipping 20 frames, with the optimal range being 20 to 40 frames.

Figure 8 presents multiple techniques for selecting the number of frames to skip, including a fixed number, random selection, and a dynamic approach using SSIM

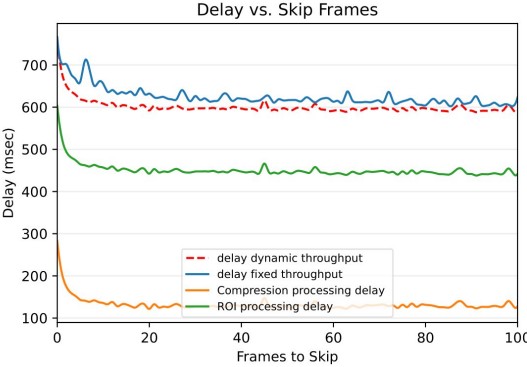

(a) Delay vs. Frames to skip

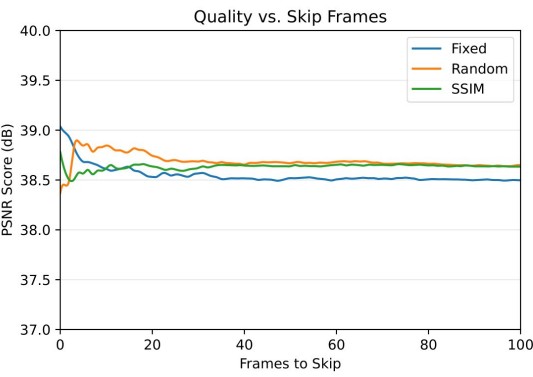

(b) Quality vs. Frames to skip

Fig. 7: Delay and Quality vs Frames to skip

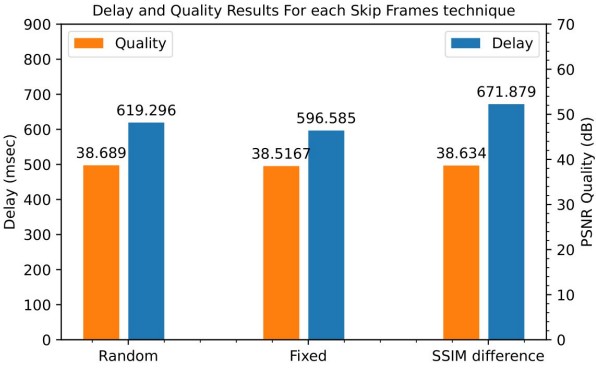

Fig. 8: Delay and quality comparison between skip frames techniques

scores. Although dynamic techniques are more adaptable, comparing frames and computing SSIM requires more resources. However, the surgeon or user has the option to select this method of operation. The user could choose a dynamic technique that adapts to the current environment but at the cost of increased delay. The

comparison among the three techniques for skipping frames, namely random, fixed, and SSIM difference, showed that all three techniques produced close quality scores, with fixed frame skipping outperforming the other two in total delay. Specifically, the total delay was 619.30 ms and 671.88 ms for the random and SSIM difference techniques, respectively. In comparison, the fixed technique reduced the total delay to 596.59 ms, a reduction of around 3.7% and 11.2%, respectively. In terms of quality, all three techniques produced similar results, with a quality score of around 38.5 dB to 38.7 dB. Therefore, while all three techniques produced comparable quality, the fixed technique is the most efficient option for reducing processing time and achieving low latency in video processing. By leveraging this technique in the system, the average total delay was decreased by approximately 18%.

## IV. Conclusion and Future Work

This paper proposes a novel system and techniques integrating robotics, virtual reality, and machine learning technologies for controlling and streaming endoscopic operations. It was demonstrated that head motion from HoloLens IMU data could prevent the endoscope camera's robotic arm. Also, the DRL model can reduce wireless video stream latency using the adaptive ROI technique. We can conclude that the fixed number skip frame technique was the most effective method for lowering video processing time and achieving low latency. These contributions and findings substantially impact the development of intelligent, efficient, and readily controlled endoscopic surgical operations.

For future work, adding predefined speech commands to control the camera or the quality of could enhance the system. Also, an automatic discovery and navigation option for surgeons will be helpful and vital for them to reach and detect some common disease markers. Finally, exploring the expansion of this work in other medical procedures such as Esophagectomy, Laparoscopic Surgery, and da Vinci Surgical Robotic-Assisted Surgery.

## Acknowledgements

This work was supported by an NPRP award (NPRP13S-0205-200265) from the Qatar National Research Fund (a member of The Qatar Foundation). The work of Abdelrahman Soliman was also supported by the GSRA award (GSRA10-L-2-0604-23091) from the Qatar National Research Fund (a member of The Qatar Foundation. The findings achieved herein are solely the responsibility of the authors.

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
