# OpenReview forum: "DRL-based low Latency control for Endoscopic operations"
_IEEE.org/EMBS/BHI/2024/Conference — IEEE BHI'24_

### Official Review · Reviewer_iHYn · 2024-08-10
**Review of 'DRL-based Low Latency Control for Endoscopic Operations'**

**Overall Rating:** 7
**Confidence:** 4

**Other Quality Metrics:**

Clarity of Writing: Great.
Clinical Significance: Excellent.
Methodological Novelty: Good.
Experiments and Results: Fair.

**Questions For The Authors:**

can you give some explainations that in fig 5, why training acc is jumping from 0.8 to 0.92 at very beginging epoch while testing accuracy is always around 0.92?
Now, what is the biggest changellge to apply this in real word ?

**Strengths:**

Combining DRL with augmented reality (AR) for controlling robotic arms in surgical operations is highly innovative, addressing key issues in the field such as latency and precision.
The system was rigorously tested in a simulation environment, providing robust proof of concept and demonstrating practical enhancements in surgical settings
The research addresses a critical need in surgical

**Summary Of The Paper:**

The paper proposes a system that uses head motion data from a HoloLens device to control a robotic arm for endoscopic operations, enhancing the control and precision during minimally invasive surgeries. A key innovation is the application of a Deep Reinforcement Learning (DRL) model to manage the region of interest in the video feed, which improves the surgeon's interaction by adaptively managing video quality and latency based on wireless channel conditions. The system was tested in a simulated environment, showing significant improvements in communication delay and image quality.

**Weaknesses:**

While simulation results are promising, real-world application and testing in actual surgical environments are necessary to fully validate the effiency and safety of the findings.

Considering the potential interference from other electronic devices in a surgical setting, such as WiFi and cellular signals, is essential. A detailed analysis of how such noise impacts the system's performance could enhance the robustness and reliability of the method, making it more applicable in diverse clinical scenarios.

---

> ### Author Rebuttal · Authors · 2024-09-01
>
> Dear Reviewers,
>
> We sincerely thank the reviewers for their time and valuable feedback on our work. We appreciate the opportunity to address their comments and improve our work. Below, we respond to each point raised:
>
> Reviewer Comment 1: While simulation results are promising, real-world application and testing in actual surgical environments are necessary to validate the efficiency and safety of the findings fully.
>
> We appreciate the reviewer highlighting this crucial aspect of the real world. The real-world demo would help us evaluate, validate, and present the utility of this approach, and we want to inform you that some progress is being made to shift the simulation to real-world implementation.
>
> Reviewer Comment 2: Considering the potential interference from other electronic devices in a surgical setting, such as WiFi and cellular signals, is essential. A detailed analysis of how such noise impacts the system's performance could enhance the robustness and reliability of the method, making it more applicable in diverse clinical scenarios.
>
> Thank you for bringing up this point. We already plan to do channel modulation and noise evaluation to assess the system's performance. As stated, it would be good to add it to this work, but we will consider it in future publications due to limited time.
>
>
>
> Reviewer Comment 3: Can you give some explanations why, in Fig 5, training acc is jumping from 0.8 to 0.92 at the very beginning epoch while testing accuracy is always around 0.92? Now, what is the biggest changelog to applying this in the real world?
>
>
> - Due to the usage of basic S-CNN, which is more than enough to find the ROI. The jump from 0.8 to 0.92 in training accuracy at the beginning is because the model rapidly adapted to the ROI detection task, thanks to a fast convergence rate after using the Adam optimizer.
>
> The biggest challenge for real-world application is ensuring the system's reliability and safety under varying surgical conditions. This includes:
> a) A very complex Mapping of the IMU sensor in the HoloLens and the control parameters needs to be presented to prevent any unwanted or unexpected outputs
> b) The medical application usually needs to be proven clinically and should be placed under several examinations by specialists.
>
>
>
> Sincerely,
> Authors team

---

### Official Review · Reviewer_Y4CP · 2024-08-10

**Overall Rating:** 7
**Confidence:** 1

**Other Quality Metrics:**

(a) Clarity of writing: good
(b) Clinical Significance: greaat
(c) Methodological Novelty: great
(d) Experiments and Results: great

**Questions For The Authors:**

1. Can the method be expanded to incorporate other types of surgeries or medical procedures beyond endoscopy?

**Strengths:**

1. The paper presents a novel integration of DRL with augmented reality (HoloLens) to improve endoscopic operations, which is a significant contribution to the field of minimally invasive surgery.
2. The system effectively reduces communication delay and improves image quality, which are critical factors in endoscopic surgeries.

**Summary Of The Paper:**

This study introduces an intelligent system for endoscopic camera control using HoloLens and Deep Reinforcement Learning (DRL) to enhance precision and reduce latency. The proof-of-concept shows the potential for advanced machine learning and augmented reality to improve endoscopic surgery.

**Weaknesses:**

1. The paper assumes the availability of high computational resources, which may not be feasible in all clinical settings.

---

> ### Author Rebuttal · Authors · 2024-09-01
>
> Dear Reviewers,
>
> We sincerely thank the reviewers for their time and valuable feedback on our work. We appreciate the opportunity to address their comments and improve our work. Below, we respond to each point raised:
>
> Reviewer Comment 1: The paper assumes the availability of high computational resources, which may only be feasible in some clinical settings.
>
> - We thank the reviewer for highlighting this valuable point. We acknowledge that the computational requirements of our method may be a limitation in some clinical settings. To address this:
>
> - In the subsection "Optimizing processing time," we tried to further optimize the processing time by introducing an efficient technique necessary to maximize video processing for low-cost and low-latency streaming to minimize the recurrent prediction of the region of interest (ROI).
>
>
> Reviewer Comment 2: Can the method be expanded to incorporate other surgeries or medical procedures beyond endoscopy?
>
> - We appreciate this insightful question about the broader applicability of our method. While our current study focused on endoscopy, our approach could expand to other medical procedures. The three main components that are needed in the different procedures are robot-assisted control, live streaming, and manual control
>
> - We will add some examples in the Future work section to highlight the possible other types that our approach can extend, such as Esophagectomy, Laparoscopic Surgery, and da Vinci Surgical Robotic-Assisted Surgery
>
> These revisions will significantly strengthen our manuscript and address the reviewers' concerns. We look forward to your feedback and to the opportunity to submit a revised version of our work.
>
> Sincerely,
> Authors team

---

### Decision · Program_Chairs · 2024-09-23

Accept